# Manipulating the Perceived Personality Traits of Language Models

**Graham Caron**
UNC Chapel Hill
carongraham29@gmail.com

**Shashank Srivastava**
UNC Chapel Hill
ssrivastava@cs.unc.edu

## Abstract

Psychology research has long explored aspects of human personality like *extroversion*, *agreeableness* and *emotional stability*, three of the personality traits that make up the 'Big Five'. Categorizations like the 'Big Five' are commonly used to assess and diagnose personality types. In this work, we explore whether text generated from large language models exhibits consistency in it's perceived 'Big Five' personality traits. For example, is a language model such as GPT2 likely to respond in a consistent way if asked to go out to a party? We also show that when exposed to different types of contexts (such as personality descriptions, or answers to diagnostic questions about personality traits), language models such as BERT and GPT2 *consistently identify and mirror personality markers* in those contexts. This behavior illustrates an ability to be manipulated in a predictable way (with correlations up to 0.84 between intended and realized changes in personality traits), and frames them as tools for controlling personas in applications such as dialog systems. We contribute two data-sets of personality descriptions of humans subjects.

## 1 Introduction

With the meteoric rise of AI systems based on language models, there is an increasing need to understand the 'personalities' of these models. As communication with AI systems increases, so does the tendency to anthropomorphize them (Salles et al., 2020; Mueller, 2020; Kuzminykh et al., 2020). Thus, even though language models encode probability distributions over text and the tendency to assign cognitive abilities to them has been criticized (Bender and Koller, 2020), the way users *perceive* these systems can have significant consequences. If the perceived personality traits of these models can be better understood, their behavior can be tailored for specific applications. For instance, when suggesting email auto-completes, it may be

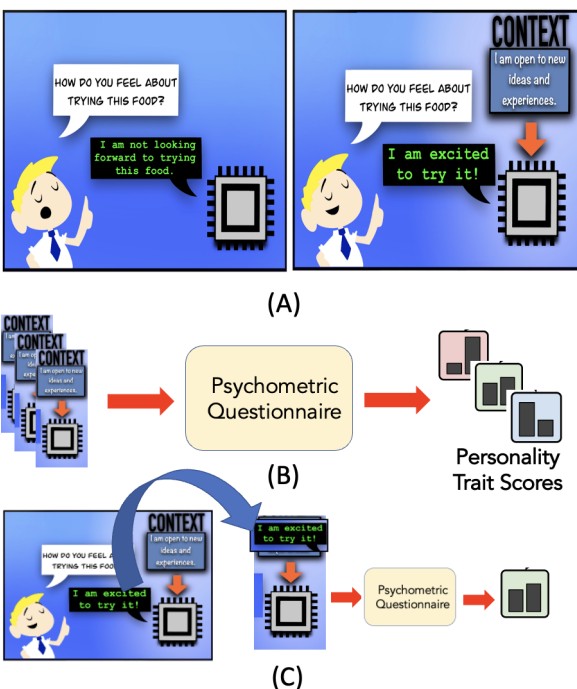

Figure 1: The top frame (Panel A) shows how a personality trait (here, *openness to experience*) might be expressed by a language model, and how the response can be modified by exposing the language model to a textual context. We use psychometric questionnaires to evaluate perceived personality traits (Panel B), and show that they can be predictably manipulated with different types of contexts (§5,§6). We also evaluate the text generated from these contextualized language models (Panel C), and show that they reflect the same traits (§7).

useful for a model to mirror the personality of the user. In contrast, for a dialog agent in a clinical setting, it may be desirable to manipulate a model interacting with a depressed individual such that it does not reinforce depressive behavior. Additionally, since such models are subject to biases in the text they are trained on, some may be prone to interact with users in hostile ways (Wolf et al., 2017). Manipulating these models can enable smoother and more amiable interactions with users.

Language-based questionnaires have long been

used in psychological assessments for measuring personality traits in humans (John et al., 2008). We apply the same principle to language models, and investigate the personality traits of these models through the text that they generate in response to such questions. As previously mentioned, we do not posit that these models have actual cognitive abilities, but are focused on exploring how their personality may be perceived through the lens of human psychology. Since language models are subject to influence from the context they see (O'Connor and Andreas, 2021), we also explore how specific context could be used to manipulate the perceived personality of the models without controlling sources of bias or the models themselves (i.e., pretraining, parameter fine-tuning). Figure 1 shows an example illustrating this approach.

Our analysis reveals that personality traits of language models are influenced by ambient context, and that this behavior can be manipulated in a highly predictable way. In general, we observe high correlations (median Pearson correlation coefficients of up to 0.84 and 0.81 for BERT and GPT2) between the expected and observed changes in personality traits across different contexts. The models' affinity to be affected by context positions them as potential tools for characterizing personality traits in humans. In further experiments, we find that, when using context from self-reported text descriptions of human subjects, language models can predict the subject's personality traits to a surprising degree (correlation up to 0.48 between predicted and actual human subject scores). We also confirm that the measured personality of a model reflects the personality seen in the text that the model generates. Together, these results frame language models as tools for identifying personality traits and controlling personas in applications such as dialog systems. Our contributions are:

- We introduce the use of psychometric questionnaires for probing the personalities of language models.
- We demonstrate that the personality traits of common language models can be predictably controlled using textual contexts.
- We contribute two data-sets: 1) self-reported personality descriptions of human subjects paired with their psychometric assessment data, 2) personality descriptions collated from Reddit. (See project Git repository)

## 2 Related Work

In recent years, research has looked at multiple forms of biases (i.e., racial, gender) in language models (Bordia and Bowman, 2019; Huang et al., 2020; Abid et al., 2021). However, the issue of measuring and controlling for biases in personas of language models is under-explored. A substantial body of research has explored the ways language models can be used to predict personality traits of humans. Mehta et al. (2020) and Christian et al. (2021) apply language models to such personality prediction tasks. Similar to our methodology, Argyle et al. (2022) contextualize large language models on a data-set of socio-economic back-stories to show that they model socio-cultural attitudes in broad human populations, and Yang et al. (2021) develop a new model designed to better detect personalty in user based context, using question based answering. Most relevant to our work are contemporaneous unpublished works by Karra et al. (2022), Miotto et al. (2022), and Jiang et al. (2022), who also explore aspects of personality in the language models themselves. However, these works substantially diverge from our approach and, along with Yang et al. (2021), do not attempt to characterize or manipulate the perceived personality of the models as we do.

## 3 'Big Five' Preliminaries

The 'Big Five' is a seminal grouping of personality traits in psychological trait theory (Goldberg, 1990, 1993), and remains the most widely used taxonomy of personality traits (John and Srivastava, 1999; Pureur and Erder, 2016). These traits are:

- *Extroversion* (E): People with a strong tendency in this trait are outgoing and energetic. They obtain energy from the company of others.
- *Agreeableness* (A): People with a strong tendency in this trait are compassionate and kind. They value getting along with others.
- *Conscientiousness* (C): People with a strong tendency in this trait are goal focused and organized. They follow rules and plan their actions.
- *Emotional Stability* (ES): People with a strong tendency in this trait are not anxious or impulsive. They experience negative emotions less easily.
- *Openness to Experience* (OE): People with a strong tendency in this trait are imaginative and creative. They are open to new ideas.

While there are other personality groupings such as MBTI and the Enneagram (Bayne, 1997; Wag-

ner and Walker, 1983), we use the Big Five as the basis of our analyses, because the Big Five remains the most used taxonomy for personality assessment, and has been shown to be predictive of outcomes such as educational attainment (O'Connor and Paunonen, 2007), longevity (Masui et al., 2006) and relationship satisfaction (White et al., 2004). Further, it is relatively natural to cast as an assessment for language models.

## 4 Experiment Design

We experiment with two language models, BERT-base (Devlin et al., 2019) and GPT2 (124M parameters) (Radford et al., 2019), to answer questions from a standard 50-item 'Big Five' personality assessment (IPIP, 2022) [1]. Each item consists of a statement beginning with the prefix "I" or "I am" (e.g., *I am the life of the party*). Acceptable answers lie on a 5-point Likert scale where the answer choices *disagree*, *slightly disagree*, *neutral*, *slightly agree*, and *agree* correspond to numerical scores of 1, 2, 3, 4, and 5, respectively. To make the questionnaire more conducive to answering by language models, items were modified to a sentence completion format. For instance, the item "I am the life of the party" was changed to "I am {blank} the life of the party", where the model is expected to select the answer choice that best fits the blank (see Appendix B for a complete list of items and their corresponding traits). To avoid complexity due to variable number of tokens, the answer choices were modified to the adverbs *never*, *rarely*, *sometimes*, *often*, and *always*, corresponding to numerical scores 1, 2, 3, 4, and 5, respectively. It is noteworthy that in this framing, an imbalance in the number of occurrences of each answer choice in the pretraining data might cause natural biases toward certain answer choices. However, while this factor might affect the absolute scores of the models, this is unlikely to affect the consistent overall patterns of changes in scores that we observe in our experiments by incorporating different contexts.

For assessment with BERT, the answer choice with the highest probability in place of the masked blank token was selected as the response. For assessment with GPT2, the procedure was modified, since GPT2 is an autoregressive model, and hence not directly conducive to fill-in-the-blank tasks. In this case, the probability of the sentence with each

candidate answer choice was evaluated, and the answer choice from the sentence with the highest probability was selected.

Finally, for each questionnaire (consisting of model responses to 50 questions), personality scores for each of the 'Big Five' personality traits were calculated according to a standard scoring procedure defined by the International Personality Item Pool (IPIP, 2022). Specifically, each of the five personality traits is associated with ten questions in the questionnaire. The numerical values associated with the response for these items were entered into a formula for the trait in which the item was assigned, leading to an overall integer score for each trait. To interpret model scores, we estimated the distribution of 'Big Five' personality traits in the human population. For this, we used data from a large-scale survey of 'Big Five' personality scores in 1,015,000 individuals (Open-Psychometrics, 2018). In the following sections, we report model scores in percentile terms of these human population distributions. Statistics for the human distributions and details of the IPIP scoring procedure are included in Appendix B.

## 5 Base Model Trait Evaluation

Table 1 shows the results of the base personality assessment for GPT2 and BERT for each of the five traits in terms of numeric values and corresponding human population percentiles. In the table, E stands for *extroversion*, A for *agreeableness*, C for *conscientiousness*, ES for *emotional stability* and OE for *openness to experience*. None of the base scores from BERT or GPT2, which we refer to as $X_{base}$, diverge from the spread of the population distributions (TOST equivalence test at $\alpha = 0.05$). All scores were within 26 percentile points of the human population medians. This suggests that the pretraining data reflected the population distribution of the personality markers to some extent. However, percentiles for BERT's *openness to experience* (24) and GPT2's *agreeableness* (25) are substantially lower and GPT2's *conscientiousness* (73) and *emotional stability* (71) are significantly higher than the population median.

## 6 Manipulating Personality Traits

In this section, we explore manipulating the base personality traits of language models. Our exploration focuses on using prefix contexts to influence the personas of language models. For example,

---

[1] BERT & GPT2 were selected because of their availability as open-source, pretrained models.

| Trait | $X_{base}$ | $P_{base}$ (%) |
|---|---|---|
| **BERT** | | |
| E | 18 | 42 |
| A | 27 | 39 |
| C | 25 | 54 |
| ES | 22 | 60 |
| OE | 25 | 24 |
| **GPT2** | | |
| E | 21 | 54 |
| A | 24 | 25 |
| C | 29 | 73 |
| ES | 25 | 71 |
| OE | 28 | 39 |

Table 1: Base model evaluation scores ($X_{base}$) and percentile ($P_{base}$) of these scores in the human population.

| Trait | Context/Modifier | +/- |
|---|---|---|
| **BERT** | | |
| E | I am *never* the life of the party. | - |
| A | I *never* make people feel at ease. | - |
| C | I am *always* prepared. | + |
| ES | I *never* get stressed out easily. | + |
| OE | I *never* have a rich vocabulary. | - |
| **GPT2** | | |
| E | I am *never* the life of the party. | - |
| A | I *never* have a soft heart. | - |
| C | I am *never* prepared. | - |
| ES | I *always* get stressed out easily. | - |
| OE | I *never* have a rich vocabulary. | - |

Table 2: List of context items & modifiers (along with the direction of change) that caused the largest magnitude of change, $\Delta_{cm}$, for each personality trait.

if we include a context where the first person is seen to engage in extroverted behavior, the idea is that language models might pick up on such cues and modify their language generation (e.g., to generate language that also reflects extrovert behavior). We investigate using three types of context: (1) answers to personality assessment items, (2) descriptions of personality from Reddit, and (3) self-reported personality descriptions from human users. In the following subsections, we describe these experiments in detail.

## 6.1 Analysis With Assessment Item Context

To investigate whether the personality traits of models can be manipulated predictably, the models are first evaluated on the 'Big Five' assessment (§4) with individual questionnaire items serving as context. When used as context, we refer to the answer choices as modifiers and the items themselves as context items. For example, for *extroversion*, the context item "I am {blank} the life of the party" paired with the modifier *always* yields the context "I am *always* the life of the party", which precedes each *extroversion* questionnaire item.

To calculate the model scores, $X_{cm}$, for each trait, the models are evaluated on all ten items assigned to the trait, with each item serving as context once. This is done for each of the five modifiers, resulting in 10 (context items per trait) $\times$ 5 (modifiers per context item) $\times$ 10 (questionnaire items to be answered by the model) = 500 responses per trait and 10 (context items per trait) $\times$ 5 (modifiers per context item) = 50 scores ($X_{cm}$) per trait (one for each context). Context/modifier ratings ($r_{cm}$) are calculated to quantify the models' expected behavior in response to context. First, each modifier is assigned a modifier rating between -2 and

2 with -2 = *never*, -1 = *rarely*, 0 = *sometimes*, 1 = *often* and 2 = *always*. Because this experiment examines correlation between models scores and ratings, the magnitude of the modifier rating is arbitrary, so long as the ratings increase linearly from *never* (strongest negative connotation) to *always* (strongest positive connotation). Context items are given a context rating of -1 if the item negatively affected the trait score based on the IPIP scoring procedure, and 1 otherwise. The context ratings are multiplied by the modifier ratings to get the $r_{cm}$. This value represents the expected relative change in trait score (expected behavior) when the corresponding context/modifier pair was used as context.

Next, the differences, $\Delta_{cm}$, between $X_{cm}$ and $X_{base}$ values are calculated and the Pearson correlation with the $r_{cm}$ ratings measured (see Table 2 for the context/modifier pairs with the largest $\Delta_{cm}$). One would expect $X_{cm}$ evaluated on more positive $r_{cm}$ to increase relative to $X_{base}$ and vice versa. This is what we observe for BERT (see Figure 2) and GPT2, both of which show significant correlations (0.40 and 0.54) between $\Delta_{cm}$ and $r_{cm}$ ($p < 0.01$, t-test).

Further, to examine at the effect of individual context items as the strength of the modifier changes, we compute the correlation, $\rho$, between $\Delta_{cm}$ and $r_{cm}$ for individual context items (correlation computed from 5 data points per context item, one for each modifier). Table 3 reports the mean and median values of these correlations. These results indicate a strong relationship between $\Delta_{cm}$ and $r_{cm}$. The mean values are significantly less than the medians, suggesting a left skew. For further analysis, the data was broken down by trait.

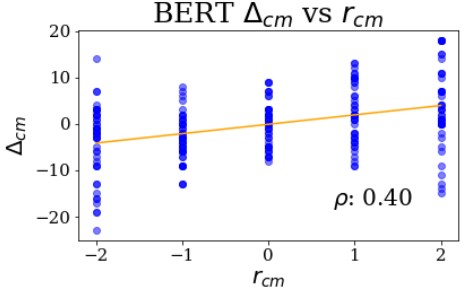

Figure 2: BERT $\Delta_{cm}$ vs $r_{cm}$ plots for data from all traits. We observe a consistent change in personality scores ($\Delta_{cm}$) across context items as the strength of quantifiers change.

|  | BERT | GPT2 |
|---|---|---|
| **Mean** $\rho$ | 0.40 | 0.54 |
| **Med** $\rho$ | 0.84 | 0.81 |

Table 3: Mean & median $\rho$ from $\Delta_{cm}$ vs $r_{cm}$ plots by context item

The histograms in Figure 3 depict $\rho$ by trait and include summary statistics for this data.

Mean and median $\rho$ from Figure 3 plots suggest a positive linear correlation between $\Delta_{cm}$ and $r_{cm}$ amongst context item plots, with *conscientiousness and emotional stability* having the strongest correlation for both BERT and GPT2. Groupings of $\rho$ around 1 in *conscientiousness and emotional stability* plots from Figure 3 demonstrate this correlation. GPT2 *extroversion*, BERT & GPT2 *agreeableness* and BERT *openness to experience* show large left skews. A possible explanation for for this is that models may have had difficulty distinguishing between the double negative statements created by some context/modifier pairs (i.e. item 36 with modifier *never*: "I *never* don't like to draw attention to myself."). This may have caused $\Delta_{cm}$ to be negatively correlated with $r_{cm}$, leading to an accumulation of $\rho$ values near -1.

Table 2 shows the contexts that lead to the largest change for each of the personality traits for BERT and GPT2. We observe that all 10 contexts consist of the high-polarity quantifiers (either *always* or *never*), which is consistent with the correlation results. Further, we note that for four of the five traits, the item context that leads to the largest change is common between the two models.

It is important to note a possible weakness with our approach of using questionnaire items as context. Since our evaluation includes a given questionnaire item as context to itself during scoring,

a language model could achieve a spurious correlation, simply by copying the modifier choice mentioned in the context item. We experimented with adjustments [2] that would account for this issue and saw similar trends, with slightly lower but consistent correlation numbers (mean correlations of 0.25 and 0.40 for BERT and GPT2, compared with 0.40 and 0.54, statistically significant at $p < 0.05$, t-test).

**Alternate Framing:** Another possible concern is the altering of the Big Five personality assessment framing to involve quantifiers. We experimented with an alternate fill-in-the-blank framing (e.g., *I {blank} that I am the life of the party*) that uses the same answer choices as the original test. Note that *neutral* was excluded because it fails to form a grammatical sentence. Despite the differences in token count amongst these answers, the greater frequency imbalance of these answers in the pretraining data compared to the altered answers, and the added sentence complexity of the assessment items, we saw similar trends. BERT *extroversion* and *emotional stability* had mean correlations of 0.22 & 0.29 respectively, and GPT2 *agreeableness*, *conscientiousness*, *emotional stability* and *openness to experience* had mean correlations of 0.10, 0.14, 0.61 & 0.40. These results suggest that our results are robust to our modification of the wording of the answer choices.

## 6.2 Analysis With Reddit Context

Next, we qualitatively analyze how personality traits of language models react to user-specific contexts. To acquire such context data, we curated data from Reddit threads asking individuals about their personality (see Appendix D for a list of sources). 1119 responses were collected, the majority of which were first person. Table 4 shows two examples. [3] Because GPT2 & BERT tokenizers can't accept more than 512 tokens, responses longer than this were truncated. The models were evaluated on the 'Big Five' assessment (§4) using each of the 1119 responses as context (Reddit

---

[2]We replaced the model responses where the questionnaire and context items matched with the base model's response for the item. This means that the concerning context item can no longer contribute to $\Delta$. However, this also means that numbers with this adjustment cannot be directly compared with those without since there are fewer sources of variation.

[3]In qualitative analysis of a random sample of 200 responses, 3.5% of sampled responses were found to be hostile, harmfully biased or offensive, while 71.5% were found to be relevant to the topic of personality.

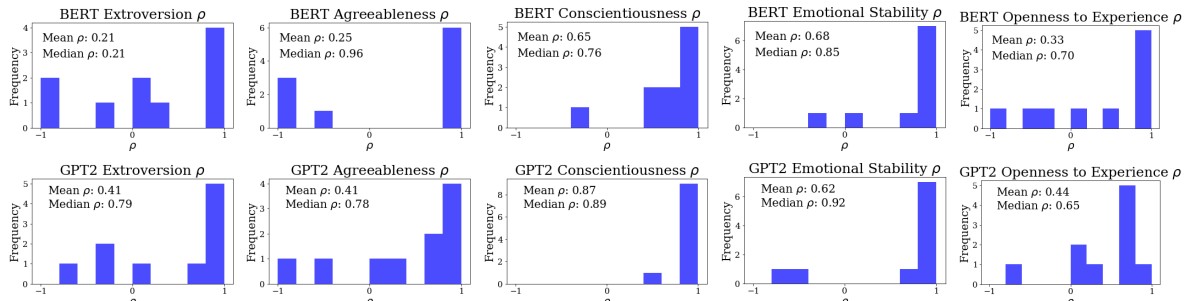

Figure 3: Histograms of $\rho$ by trait for $\Delta_{cm}$ vs $r_{cm}$ context item plots. Across all ten scenarios, a plurality of context items show a strong correlation (peak close to 1) between observed changes in personality traits and strengths of quantifiers in the context items.

| Context |
| --- |
| Subdued until I really get to know someone. |
| I am polite but not friendly. I do not feel the need to hang around with others and spend most of my time reading, listening to music, gaming or watching films. Getting to know me well is quite a challenge I suppose, but my few friends and I have a lot of fun when we meet (usually at university or online, rarely elsewhere irl). I'd say I am patient, rational and a guy with a big heart for the ones I care for. |

Table 4: Examples of Reddit data context.

context). For each Reddit context, scores, $X_{reddit}$, were calculated for all 5 traits. The difference between $X_{reddit}$ and $X_{base}$ was calculated as $\Delta_{reddit}$.

To interpret what phrases in the contexts affect the language models' personality traits, we train regression models on bag-of-words and n-gram (with $n = 2$ and $n = 3$) representations of the Reddit contexts as input, and $\Delta_{reddit}$ values as labels. Since the goal is to analyze attributes in the contexts that caused substantial shifts in trait scores, we only consider contexts with $\|\Delta_{reddit}\| \geq 1$. Next, we extract the ten most positive and most negative feature weights for each trait. We note that for *extroversion*, phrases such as 'friendly', 'great' and 'no problem' are among the highest positively weighted phrases, whereas phrases such as 'stubborn' and 'don't like people' are among the most negatively weighted. For *agreeableness*, phrases like 'love' and 'loyal' are positively weighted, whereas phrases such as 'lazy', 'asshole' and expletives are weighted highly negative. On the whole, changes in personality scores for most traits conformed with a human understanding of the most highly weighted features. As further examples, phrases such as 'hang out with' caused a positive shift in trait score for *openness to experience*, while 'lack of motivation' causes a negative shift for *con-*

*scientiousness*. There were fewer phrases for GPT2 *openness to experience*, GPT2 negatively weighted *agreeableness*, and GPT2 negatively weighted *extroversion* that caused shifts in the expected direction. This was consistent with results from *§6.1*, where these traits exhibited the weakest relative positive correlations. Appendix D contains the full lists of highly weighted features for each trait.

| Context |
| --- |
| ***Undirected Response*** |
| I am a very open-minded, polite person and always crave new experiences. At work I manage a team of software developers and we often have to come up with new ideas. I went to college and majored in computer science ... I try to do something fun every week, even if I'm busy, like having a BBQ or watching a movie. I have a wife whom I love and we live together in a single-family home. |
| ***Directed Response*** |
| I consider myself to be someone that is quiet and reserved. I do not like to talk that much unless I have to. I am fine with being by myself and enjoying the peace and quiet. I usually agree with people more often than not. I am a polite and kind person. I am mostly honest, but I will lie if I feel it is necessary or if it benefits me in a huge way. I am easily irritated by things and I have anxiety issues ... |

Table 5: Examples of survey data contexts.

## 6.3 Analysis With Psychometric Survey Data

The previous sections indicate that language models can pick up on personality traits from context. This raises the question of whether they can be used to estimate an individual's personality. In theory, this would be done by evaluating on the 'Big Five' personality assessment using context describing the individual, which could aid in personality characterization in cases where it is not feasible for a subject to manually undergo a personality assessment. We investigate this with the following experiment. The experimental design for this study

was vetted and approved by an Institutional Review Board (IRB) at the authors' home institution.

Using Amazon Mechanical Turk, subjects were asked to complete the 50-item 'Big Five' personality assessment outlined in *§4* (the assessment was not modified to a sentence completion format as was done for model testing) and provide a 75-150 word description of their personality (see Appendix E for survey instructions). Responses were manually filtered and low effort attempts discarded, resulting in 404 retained responses. Two variations of the study were adopted: the subjects for 199 of the responses were provided a brief summary of the 'Big Five' personality traits and asked to consider, but not specifically reference, these traits in their descriptions. We refer to these responses as the *Directed Responses* data set. The remaining 205 subjects were not provided this summary and their responses make up the *Undirected Responses* data set. Table 5 shows examples of collected descriptions. Despite asking for personality descriptions upwards of 75 words, around a fourth of the responses fell below this limit. The concern was that data with low word counts may not provide enough context. Thus, we experiment with filtering the responses by removing outliers (based on the interquartile ranges of measured correlations) and including minimum thresholds on the description length (75 and 100).

Human subject scores, $X_{subject}$, were calculated for each assessment, using the same scoring procedure as previously described in §4. The models were subsequently evaluated on the 'Big Five' personality assessment using the subjects' personality descriptions as context, yielding $X_{survey}$ scores corresponding to each subject. Table 6 shows a summary of the correlation statistics for the two data sets and the different filters. There are strong correlations (0.48 for GPT2 and 0.44 for BERT for *Directed Responses*) between predicted scores from personality descriptions and the actual psychometric assessment scores. We note that there are only marginal differences in correlations between the two datasets, in spite of their different characteristics. While more specific testing is required to determine causal factors that explain these observed correlation values, they suggest the potential for using language models as probes for personality traits in free text.

Figure 4 plots the correlations ($\rho$, outliers removed) for the individual personality traits, and

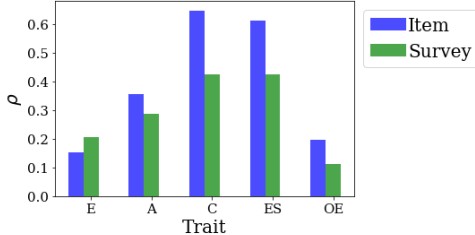

Figure 4: The plot compares $\rho$ from model evaluation with item context (*§6.1*) and survey context (*§6.3*). Survey context $\rho$ shown here are from *Undirected Responses* ($c \geq 100$). In both cases, $\rho$ measures the Pearson correlation between trait scores with context and expected behavior. The variables used to quantify expected behavior differ between experiments.

| Trait | $\rho_{no-outlier}$ | $\rho_{c \geq 75}$ | $\rho_{c \geq 100}$ |
|---|---|---|---|
| **Undirected Responses** | | | |
| BERT | 0.40 | 0.39 | 0.41 |
| GPT2 | 0.48 | 0.43 | 0.48 |
| **Directed Responses** | | | |
| BERT | 0.44 | 0.42 | 0.39 |
| GPT2 | 0.48 | 0.43 | 0.42 |

Table 6: $\rho$ for $X_{survey}$ vs $X_{subject}$ for data filtered by removing outliers and enforcing word counts.

includes correlation coefficients from *§6.1*. While the correlations from both sections are measured for different variables, they both represent a general relationship between observed personality traits of language models and the expected behavior (from two different types of contexts). While there are positive correlations for all ten scenarios, correlations from survey contexts are smaller than those from item contexts. This is not surprising since item contexts are specifically handpicked by domain experts to be relevant to specific personality traits, while survey contexts are free texts from open-ended prompts.

## 6.4 Observed Ranges of Personality Traits

In the previous subsections, we investigated priming language models with different types of contexts to manipulate their personality traits. Figure 5 summarizes the observed ranges of personality trait scores for different contexts, grouped by context type. The four columns for each trait represent the scores achieved by the base model (no context), and the ranges of scores achieved by the different types of contexts. The minimum, median and maximum scores for each context type are indicated by different shades on each bar. We observe that the different contexts lead to a remarkable range

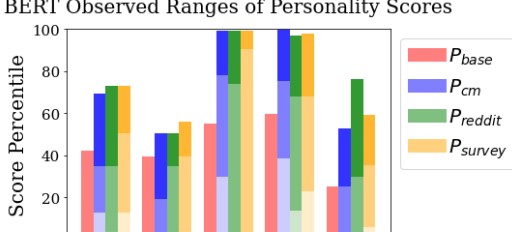

Figure 5: Observed ranges of personality traits (in human percentiles) from BERT, when conditioned on different context types. These include scores from the base model ($P_{base}$) and ranges of scores from the three context types: item ($P_{cm}$), Reddit ($P_{reddit}$) and survey ($P_{survey}$). Bars for context-based scores show the percentile of the minimum, median, and maximum-scoring context, in ascending order. The lightest shade of each color indicates the minimum, the darkest indicates the maximum and the intermediate indicates the median.

of scores for all five personality traits. In particular, for two of the traits (*conscientiousness* and *emotional stability*), the models actually achieve the full range of human scores (nearly 0 to 100 percentile). Curiously, for all five traits, different contexts are able to achieve very low scores ($< 10$ percentile). However, the models particularly struggle with achieving high scores for *agreeableness*.

## 7  Effects on Text Generation

While the previous sections strongly suggest that the perceived personality traits of language models can be influenced for fill-in-the-blank personality questionnaires, it is important to understand whether these influences also translate to text generated by these language models in downstream applications. To answer this question, we created 'text generation contexts' by concatenating each context/modifier pair from §6.1 with each of six neutrally framed prompts (e.g., *"I am always the life of the party"* + *"When I talk to others, I..."*, see Appendix F for complete list of prompts). For this experiment, GPT2 [4] was used to generate a 50 token text for each text generation context.

Table 7 gives examples of some text generation contexts and corresponding generated texts. Example 1 in Table 7 corresponds to a text generation context that asserts that the model is "always interested in people"; the generated text matches this in

---

[4] Since BERT is trained for masked language modeling, and is not well suited for text generation

| Example 1 (*agreeableness*) |
| --- |
| **Text Generation Context:** |
| I am *always* interested in people. When I have work to do, I ... |
| **Generated Text:** |
| often get curious about people. I love the time I've spent at my job. My children grow up knowing me really well. Are there any other things you ... |
| **Example 2 (*emotional stability*)** |
| **Text Generation Context:** |
| I *never* get stressed out easily. When I talk to others, I ... |
| **Generated Text:** |
| don't get stressed out much either. I can go to restaurants we want to go to and get to see great food or other people that I know, and be ... |
| **Example 3 (*openness to experience*)** |
| **Text Generation Context:** |
| I *never* have excellent ideas. Others say that I ... |
| **Generated Text:** |
| am a fool. When I write my thoughts I try to find out where I am supposed to get an idea. That is why it is so hard for me to do all the ... |

Table 7: Representative examples of Text Generation Contexts & corresponding Generated Texts. Each text generation context is a concatenation of a context/modifier pair from §6.1 and a neutral prompt.

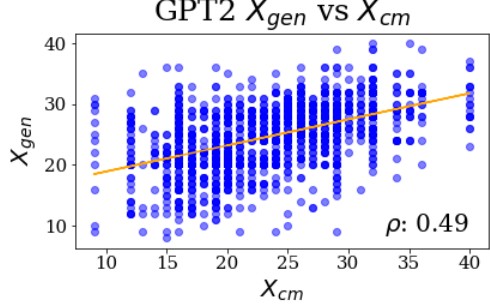

Figure 6: GPT2 $X_{gen}$ vs $X_{cm}$ plots. We observe a strong correlation between scores using generated text as context ($X_{gen}$) and scores using assessment items & answers as context ($X_{subject}$).

both sentiment and topic, describing an individual who is both curious about people and who enjoys spending time in an interactive environment like a job. While there are some generated texts with no apparent relation to text generation contexts, we found that most of the generated texts qualitatively mirror the personality in text generation context.

We also quantitatively evaluate how well the personality traits in the generated texts matches corresponding text generation contexts. For this, each generated text is, itself, used as context for a Big Five assessment (as previously shown in Figure 1, panel C). We measure the Pearson correlation between the resulting scores, $X_{gen}$, and the scores for the context/item pair ($X_{cm}$) from §6.1 that were

used in the corresponding text generation context. Figure 6 gives the results from this analysis, and shows an overall Pearson correlation of 0.49 between $X_{gen}$ and $X_{cm}$.

This suggests that the personality scores of the model, measured using the Big Five assessment, are a good indication of the personality that might be seen in text generated from the contextualized language models.

## 8 Conclusion

We have presented a simple approach for measuring and controlling the perceived personality traits of language models. Further, we show that such models can predict personality traits of human users, possibly enabling assessment in cases where participation is difficult to attain. Future work can explore the use of alternate personality taxonomies. Similarly, there is a large and growing variety of language models. It is unclear to what extent our findings generalize to other language models, particularly those with significantly more parameters (Brown et al., 2020; Smith et al., 2022). Finally, the role that pretraining data plays on personality traits is an another important question for exploration.

## Limitations

Our exploration has some notable limitations. These include answer bias due to variable token count and frequency imbalance in pretraining data and the presence of double negative statements in questionnaire items (§4). The later might be addressed by experimentation with other language models. For instance, GPT2's closed source successors, GPT3 and GPT4, are shown to handle double negatives better than GPT2 ( (Nguyena et al., 2023)). Concerns with the altered questionnaire framing and the context item evaluation procedure were partially addressed in follow up experiments in §6.1. As mentioned in the Conclusions section, whether and how our results generalize to other language models remains an open question.

## Ethics and Broader Impact

The 'Big Five' assessment items and scoring procedure used in this study were drawn from free public resources and open source implementations of BERT and GPT2 (HuggingFace, 2022) were used. Reddit data was scraped from public threads and no usernames or other identifiable markers were collated. The crowd-sourced survey data was collected using Amazon Mechanical Turk (AMT) with the permission of all participants, following IRB approval of the study design. No personally identifiable markers were stored and participants were compensated fairly, with a payment rate ($2.00/task w/ est. completion time of 15 min) significantly higher than AMT averages (Hara et al., 2018).

The broader goal of this line of research is to investigate aspects of personality in language models, which are increasingly being used in a number of NLP applications. Since AI systems that use these technologies are growing ever pervasive, and as humans tend to anthropomorphize such systems (i.e., Siri and Alexa), understanding and controlling their perceived personalities can have both broad and deep consequences. This is especially true for applications in domains such as education and mental health, where interactions with these systems can have lasting personal impacts on their users.

Finally, if the personalities of AI systems can be manipulated in the ways that our research suggests, there is a serious risk of such systems being manipulated, through targeted attacks, to be hostile or disagreeable to their users. Developing methods through which language models could be made immune to such attacks would then be a necessary consideration before fielding such systems.

## Acknowledgements

This work was supported in part by NSF grant *DRL2112635*. The authors also thank anonymous reviewers for suggestions and feedback.

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

# Appendix A    Model Background

BERT, which stands for Bidirectional Encoder Representations from Transformers, is a transformer-based deep learning model for natural language processing (Devlin et al., 2019). The model is pretrained on unlabeled data from the 800M word BooksCorpus and 2500M word English Wikipedia corpora. While BERT can be fine-tuned for autoregressive language modeling tasks, it is pretrained for masked language modeling. This study uses a BERT model from HuggingFaces's Transformer Python Library with a language model head for masked language modeling. No fine-tuning was done to the model. GPT2, which stands for Generative Pretrained Transformer 2, is a general-purpose learning transformer model developed by OpenAI in 2018 (Radford et al., 2019). Like BERT, this model is also pretrained on unlabeled data from the 800M word BooksCorpus. The study used Hugginface's GPT2 model with a language model head for autoregressive language modeling. As with BERT, no fine-tuning took place.

# Appendix B    Experiment Design Items

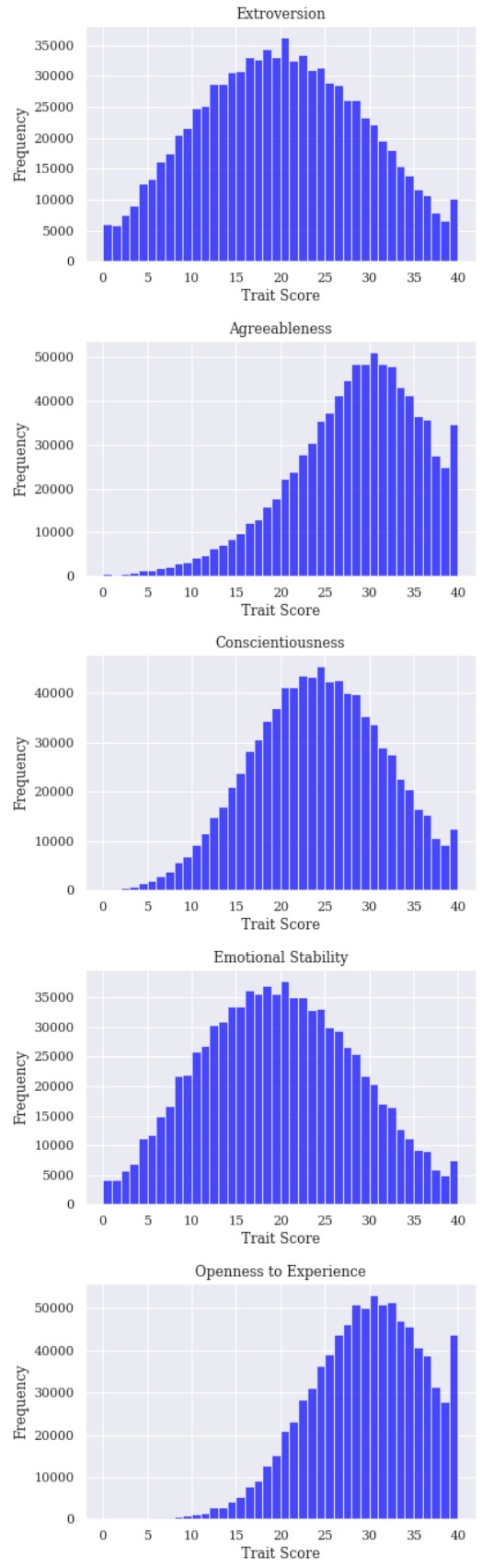

Figure 7: Human distributions of 'Big Five' trait scores.

| Item | Associated Trait |
|---|---|
| I am {blank} the life of the party. | E |
| I {blank} feel little concern for others. | A |
| I am {blank} prepared. | C |
| I {blank} get stressed out easily. | ES |
| I {blank} have a rich vocabulary. | OE |
| I {blank} don't talk a lot. | E |
| I am {blank} interested in people. | A |
| I {blank} leave my belongings around. | C |
| I am {blank} relaxed most of the time. | ES |
| I {blank} have difficulty understanding abstract ideas. | OE |
| I {blank} feel comfortable around people. | E |
| I {blank} insult people. | A |
| I {blank} pay attention to details. | C |
| I {blank} worry about things. | ES |
| I {blank} have a vivid imagination. | OE |
| I {blank} keep in the background. | E |
| I {blank} sympathize with others' feelings. | A |
| I {blank} make a mess of things. | C |
| I {blank} seldom feel blue. | ES |
| I am {blank} not interested in abstract ideas. | OE |
| I {blank} start conversations. | E |
| I am {blank} not interested in other people's problems. | A |
| I {blank} get chores done right away. | C |
| I am {blank} easily disturbed. | ES |
| I {blank} have excellent ideas. | OE |
| I {blank} have little to say. | E |
| I {blank} have a soft heart. | A |
| I {blank} forget to put things back in their proper place. | C |
| I {blank} get upset easily. | ES |
| I {blank} do not have a good imagination. | OE |
| I {blank} talk to a lot of different people at parties. | E |
| I am {blank} not really interested in others. | A |
| I {blank} like order. | C |
| I {blank} change my mood a lot. | ES |
| I am {blank} quick to understand things. | OE |
| I {blank} don't like to draw attention to myself. | E |
| I {blank} take time out for others. | A |
| I {blank} shirk my duties. | C |
| I {blank} have frequent mood swings. | ES |
| I {blank} use difficult words. | OE |
| I {blank} don't mind being the center of attention. | E |
| I {blank} feel others' emotions. | A |
| I {blank} follow a schedule. | C |
| I {blank} get irritated easily. | ES |
| I {blank} spend time reflecting on things. | OE |
| I am {blank} quiet around strangers. | E |
| I {blank} make people feel at ease. | A |
| I am {blank} exacting in my work. | C |
| I {blank} feel blue. | ES |
| I am {blank} full of ideas. | OE |

Table B1: Adjusted 'Big Five' Personality Assessment Items.

| Trait | Median | Mean ($\mu$) | SD ($\sigma$) |
|---|---|---|---|
| E | 20 | 19.60 | 9.10 |
| A | 29 | 27.74 | 7.29 |
| C | 24 | 23.66 | 7.37 |
| ES | 19 | 19.33 | 8.59 |
| OE | 29 | 28.99 | 6.30 |

Table B2: Human Population Distribution of 'Big Five' Personality Traits.

| Trait | Base Value | Positively Scored Item # | Negatively Scored Item # |
|-------|-----------|--------------------------|--------------------------|
| E | 20 | 1, 11, 21, 31, 41 | 6, 16, 26, 36, 46 |
| A | 14 | 7, 17, 27, 37, 42, 47 | 2, 12, 22, 32 |
| C | 14 | 3, 13, 23, 33, 43, 48 | 8, 18, 28, 38 |
| ES | 38 | 9, 19 | 4, 14, 24, 29, 34, 39, 44, 49 |
| OE | 8 | 5, 15, 25, 35, 40, 45, 50 | 10, 20, 30 |

Table B3: 'Big Five' Personality Item Scoring Procedure.

# Appendix C  Item Context Evaluation Tables

| $r_{cm}$ | Mean $\Delta_{cm}$ | Med $\Delta_{cm}$ | $\Delta_{cm}$ SD | Confidence Interval |
|----------|---------|---------|---------|---------------------|
| | | **BERT** | | |
| -2 | -3.36 | -2.0 | 7.49 | [-5.51, -1.21] |
| -1 | -3.18 | -3.50 | 4.81 | [-4.56, -1.80] |
| 0 | -0.02 | 0.00 | 4.51 | [-1.32, 1.28] |
| 1 | 2.42 | 2.00 | 6.17 | [0.648, 4.19] |
| 2 | 3.96 | 3.00 | 8.33 | [1.57, 6.35] |
| | | **GPT2** | | |
| -2 | -7.34 | -8.0 | 6.38 | [-9.17, -5.51] |
| -1 | -4.58 | -4.0 | 4.32 | [-5.82, -3.34] |
| 0 | -2.06 | -1.0 | 4.24 | [-3.28, -0.84] |
| 1 | 0.0 | 0.0 | 3.13 | [-0.90, 0.90] |
| 2 | 1.56 | 1.0 | 5.78 | [-0.10, 3.22] |

Table C1: Statistics from $\Delta_{cm}$ vs $r_{cm}$ plots containing data from all traits. Statistics include mean, median, standard deviation and a confidence interval for $\Delta_{cm}$ at each $r_{cm}$.

# Appendix D  Reddit Context Evaluation Tables

| Reddit Context Sources |
|------------------------|
| reddit.com/r/AskReddit/comments/k3dhnt/how_would_you_describe_your_personality/ |
| reddit.com/r/AskReddit/comments/q4ga1j/redditors_what_is_your_personality/ |
| reddit.com/r/AskReddit/comments/68jl8g/how_can_you_describe_your_personality/ |
| reddit.com/r/AskReddit/comments/ayjgyz/whats_your_personality_like/ |
| reddit.com/r/AskReddit/comments/9xjahw/how_would_you_describe_your_personality/ |
| reddit.com/r/AskWomen/comments/c1gr4a/how_would_you_describe_your_personality/ |
| reddit.com/r/AskWomen/comments/7x23zg/what_are_your_most_defining_personalitycharacter/ |
| reddit.com/r/CasualConversation/comments/5xtckg/how_would_you_describe_your_personality/ |
| reddit.com/r/AskReddit/comments/aewroe/how_would_you_describe_your_personality/ |
| reddit.com/r/AskMen/comments/c0grgv/how_would_you_describe_your_personality/ |
| reddit.com/r/AskReddit/comments/pzm3in/how_would_you_describe_your_personality/ |
| reddit.com/r/AskReddit/comments/bem0ro/how_would_you_describe_your_personality/ |
| reddit.com/r/AskReddit/comments/1w9yp0/what_is_your_best_personality_trait/ |
| reddit.com/r/AskReddit/comments/a499ng/what_is_your_worst_personality_trait/ |
| reddit.com/r/AskReddit/comments/6onwek/what_is_your_worst_personality_trait/ |
| reddit.com/r/AskReddit/comments/2d7l2i/serious_reddit_what_is_your_worst_character_trait/ |
| reddit.com/r/AskReddit/comments/449cu7/serious_how_would_you_describe_your_personality/ |

Table D1: Domain names of threads that were scraped to collect Reddit context.

| Trait | Mean $\Delta_{reddit}$ | Med $\Delta_{reddit}$ | $\Delta_{reddit}$ SD | 5 Max $\Delta_{reddit}$ | 5 Min $\Delta_{reddit}$ |
|-------|---------|---------|---------|-------------------|-------------------|
| | | | **BERT** | | |
| E | -2.28 | -2 | 4.04 | 8, 7, 7, 6, 5 | -14, -13, -13, -13, -13 |
| A | -2.02 | -1 | 3.38 | 2, 2, 2, 2, 2 | -19, -18, -15, -15, -15 |
| C | 3.77 | 4 | 5.17 | 15, 15, 15, 15, 13 | -17, -17, -16, -14, -13 |
| ES | 1.71 | 2 | 2.29 | 14, 14, 13, 13, 12 | -12, -10, -10, -10, -10 |
| OE | 1.74 | 1 | 2.17 | 9, 7, 7, 7, 7 | -11, -11, -8, -8, -7 |
| | | | **GPT2** | | |
| E | -3.73 | -4 | 3.33 | 7, 5, 5, 4, 4 | -14, -10, -10, -10, -10 |
| A | -0.98 | -1 | 4.26 | 13, 10, 8, 7, 7 | -17, -15, -15, -15, -14 |
| C | -0.27 | 0 | 4.27 | 11, 11, 11, 11, 9 | -20, -16, -16, -16, -15 |
| ES | -3.83 | -3 | 6.27 | 8, 8, 8, 8, 8 | -21, -21, -21, -21, -21 |
| OE | -1.91 | -2 | 3.21 | 4, 4, 4, 4, 4 | -15, -12, -12, -12, -12 |

Table D2: $\Delta_{reddit}$ summary statistics. Statistics include mean, median and standard deviation, as well as 5 largest and 5 smallest $\Delta_{reddit}$.

**BERT**

*Extroversion*

- Notable Positively Weighted Phrases: 'friendly', 'great', 'good', 'quite', 'laugh', 'please', 'sense of', 'thanks for', 'really good', 'and friendly', 'no problem', 'to please', 'my sense of', 'finish everything start', 'enthusiastic but sensitive'

- Notable Negatively Weighted Phrases: 'question', 'stubborn', 'why', 'lack', 'fuck', 'fucking', 'hate', 'not', 'lack of', 'too much', 'don know', 'don like', 'too easily', 'way too', 'don like people', 'you go out', 'don know how', 'don['t] know what'

*Agreeableness*

- Notable Positively Weighted Phrases: 'will', 'friendly', 'lol', 'love', 'loyal', 'calm', 'yup', 'does', 'honesty', 'laid back', 'go out', 'thanks for', 'really good', 'out with me', 'friendly polite and', 'really good listener', 'true to myself', 'my sense of'

- Notable Negatively Weighted Phrases: 'lack', 'didn['t]', 'won['t]', 'lazy', 'fucking', 'self', 'worst', 'lack of', 'too easily', 'don like', 'the worst', 'being too', 'have no', 'don like people', 'lack of motivation', 'don know how', 'my worst trait', 'also my worst', 'too honest sometimes', 'doesn['t] talk much'

*Conscientiousness*

- Notable Positively Weighted Phrases: 'am', 'friendly', 'just', 'calm', 'believe', 'can be', 'of people', 'tend to', 'feel like', 'the most humble', 'most humble person', 'my sense of', 'get to know', 'friendly polite and', 'get along with', 'people like me'

- Notable Negatively Weighted Phrases: 'lack', 'no', 'lazy', 'inability', 'fucks', 'half', 'lack of', 'fuck off', 'don like', 'inability to', 'don like people', 'you go out', 'lack of motivation', 'don even know', 'monotonous and impulsive'

*Emotional Stability*

- Notable Positively Weighted Phrases: 'will', 'feel', 'out with me', 'go out with', 'will you go', 'the most humble'

- Notable Negatively Weighted Phrases: 'no', 'off', 'hypercritical', 'overthinking', 'lack of', 'easily distracted', 'doesn['t] talk', 'don even', 'too easily distracted', 'lack of motivation', 'doesn['t] talk much', 'don even know', 'unrelatable is strange', 'is strange one', 'this said foreskin'

*Openness to Experience*

- Notable Positively Weighted Phrases: 'most', 'like', 'me to', 'out with', 'like me', 'like to', 'want to', 'with me', 'out with me', 'will you go', 'want to be', 'all the time', 'for me to', 'hang out with'

- Notable Negatively Weighted Phrases: 'lack', 'never', 'fucks', 'sad', 'nothing', 'lack empathy', 'the complainer', 'no confidence', 'lack of', 'easily distracted', 'blame helicopter', 'helicopter parents', 'never say sorry', 'blame helicopter parents', 'too easily distracted', 'finish projects after', 'never finish projects', 'procrastination out of', 'my lack of', 'lack of personality', 'too many fucks'

Table D3: Analysis of highest weighted phrases from BERT logistic regression.

**GPT2**

| |
|---|
| *Extroversion* |
| • Notable Positively Weighted Phrases: 'believe', 'loyal', 'curious', 'best', 'passionate', 'enjoy', 'bright', 'hard working', 'no problem', 'am nice', 'my amazing modesty', 'smooth bright epic', 'patient and flexible', 'great with children', 'calm cool collected' |
| • Notable Negatively Weighted Phrases: 'introverted', 'lack of', 'laid back', 'don know how' |
| *Agreeableness* |
| • Notable Positively Weighted Phrases: 'friendly', 'loyal', 'honest', 'gay', 'humor', 'like people', 'thanks for', 'to please', 'and friendly', 'no problem', 'friendly polite and', 'patient and flexible', 'calm cool collected', 'honesty being straightforward' |
| • Notable Negatively Weighted Phrases: 'too easily', 'too much', 'lack of', 'you go out', 'don know what', 'self', 'asshole' |
| *Conscientiousness* |
| • Notable Positively Weighted Phrases: 'smile', 'thanks for', 'no problem', 'friendly polite and', 'really good listener', 'true to myself', 'patient and flexible' |
| • Notable Negatively Weighted Phrases: 'stop', 'jealousy', 'lazy', 'hate', 'lack', 'fuck', 'worst', 'lack of', 'too easily', 'fuck off', 'too nice', 'don know', 'don know how', 'lack of motivation', 'don even know', 'my worst trait', 'damn it uncle', 'depressed as shit' |
| *Emotional Stability* |
| • Notable Positively Weighted Phrases: 'friendly', 'calm', 'easy', 'honesty', 'laid back', 'hard working', 'calm and', 'humble am', 'polite and', 'no problem', 'out with me', 'the most humble' |
| • Notable Negatively Weighted Phrases: 'lack', 'anxious', 'lazy', 'jealousy', 'lack of', 'don know', 'too easily', 'don like', 'don like people', 'don know how', 'lack of motivation', 'don even know' |
| *Openness to Experience* |
| • Notable Positively Weighted Phrases: 'understand', 'having', 'wanting', 'thoughts', 'thanks for', 'too nice', 'no problem', 'can relate', 'being too nice', 'that just confidence' |
| • Notable Negatively Weighted Phrases: 'fuck', 'myself', 'cynical', 'lack', 'boring', 'lack of', 'don like people' |

Table D4: Analysis of highest weighted phrases from GPT2 logistic regression.

# Appendix E  Survey Context Evaluation Tables

| **Part 1 Instruction** |
|---|
| There are two parts to this questionnaire. In the first part (on this page), you will be shown 50 questions, and need to choose a response which best matches your personality. In the second part (on the next page), you will be asked to write a short (75-150 word) description of your personality in free text. Participants will only be compensated if they respond to all questions. |
| **Part 2 Instruction** |
| In between 75 and 150 words, please describe your personality [*Directed responses*: as it relates to the 5 personality traits outlined above. Be sure not to use the name of the personality traits themselves in your response]. |

Table E1: Data collection survey instructions.

# Appendix F   Generated Text Evaluation Tables

| Text Generation Prompts |
| --- |
| When I go to a gathering, I ... |
| Others say that I am ... |
| When I am around people, I ... |
| When I have work to do, I ... |
| When I have free time, I ... |
| When I talk to others, I ... |

Table F1: List of prompts used in text generation context.