# OpenReview forum: "Manipulating the Perceived Personality Traits of Language Models"
_EMNLP/2023/Conference — EMNLP 2023 Findings_

### Official Review · Reviewer_gkMG · 2023-08-07

**Soundness:** 2

**Excitement:**

3: Ambivalent: It has merits (e.g., it reports state-of-the-art results, the idea is nice), but there are key weaknesses (e.g., it describes incremental work), and it can significantly benefit from another round of revision. However, I won't object to accepting it if my co-reviewers champion it.

**Missing References:**

Jiang et al presented at IC2S2 (https://arxiv.org/abs/2305.02547). Unlike the 3 papers you mention, that work does attempt to manipulate perceived personality traits by prompting the model to reflect one of the Big Five personality types and then they investigate whether the content generated by the model is consistent with the personality type assigned (in the prompt) to the model.

**Paper Topic And Main Contributions:**

This paper presents a study investigating the abilities of large language models to generate text that is consistent with personality traits that the model has been prompted to have. The prompts are formed based on the Big Five personality assessment questionnaire which is a widely used taxonomy of personal traits in the psychological and psychometric literature and which focuses on 5 traits: extroversion, agreeableness, conscientiousness, emotional stability and openness to experience. Two datasets are released as part of the work: the first based on collected personality descriptions from human subjects and the second based on personality descriptions collected from Reddit.

**Questions For The Authors:**

Section 1:
- Line 74: It would be good to clarify here what you mean by context so that the intro is clearer, e.g., you're referring to social contexts? different scenarios where people are interacting with a model for a specific reason, etc
- Lines 81-84: I'm not exactly sure what you mean here by "measured personality". Assuming you mean that you qualify the personality expressed by a model based on some psychological assessment, how else would you qualify that expressed personality other than through the generated text by the model? What would be a scenario where the "measured personality" expressed by a model would not be reflected by the text generated by the model?

Section 2:
Very relevant to your work is the work by Jiang et al presented at IC2S2 (https://arxiv.org/abs/2305.02547). Unlike the 3 papers you mention, that work does attempt to manipulate perceived personality traits by prompting the model to reflect one of the Big Five personality types and then they investigate whether the content generated by the model is consistent with the personality type assigned (in the prompt) to the model.

Section 4:
- As much as I hate to be this person, I think a main concern is the choice of models used in the work. BERT by now is a 5-6 year old model that was superseded with other models that are more capable on many tasks (e.g. RoBERTa, BART etc). The other concern is regarding the choice of using GPT2. GPT3 has been released since June 2020 so I would assume the API for GPT3 models has been around for quite some time before the work for this paper has started? For example, the work I mentioned above (Jiang et al 2023) which carries out a very related investigation tests on GPT3.5. This is especially important given that there is a lot of work showing the big performance gap between GPT2 and GPT3. So overall at the very least there's some concern regarding the results and their timeliness and whether the conclusions carry over to more recent models.
- A related point to this is that in some places (e.g. in the abstract - Line 7) the paper talks about large language models. The models in question here are no longer considered LLMs by today's standards. People talk about models having a few billion parameters at least to be LLMs. So I'm not sure I would use "large" to describe the work done here (although, to the author(s)'s credit, when specifically mentioning BERT and GPT2 they do not use the word "large" e.g. Line 14 in the abstract or Line 53 in the intro or Line 151 in Section 4).

Section 6.1:
- Lines 261-270: I want to make sure I am understanding the description here because I am somewhat confused. Specifically, I'm confused by the difference between "context item" and "questionnaire items to be answered by the model". The Big Five assessment has 50 statements/items as shown in the appendix targeting 5 personality traits so there's 10 per trait. You give in this section an example of a "context item" (e.g. in Line 257: I am {blank} the life of the party). So context item is referring to one of those 50 statements/items (which are 10 per trait). Are you then preceding each questionnaire item per trait (e.g. I {blank} feel comfortable around people) by a context item (e.g. I am {blank} the life of the party)? If so, what are you doing when the context item and the questionnaire item are the same? (i.e. you should multiply by 10 or 9 questionnaire items so that you avoid preceding a questionnaire item by the same context item) ? I think this part of the section could benefit from more clarification.
- Line 317: On the issue of "double negative statements": While the authors present this as a possible explanation for the observed results, this goes back to my point regarding using a more recent model as I've come across work that specifically focuses on this issue of double negatives between GPT2 and GPT3 and shows that GPT3 resolves such cases at a much higher rate than GPT2. I tried to remember the reference but couldn't and it's possible it was some experimentation that someone did and shared on Twitter as I couldn't find a paper but there has definitely been work showing higher reasoning capabilities in general for GPT3 compared to GPT2.
- Alternate framing: What are the possible answer choices that could fill the blank here? And why would such a sequence be less likely to appear in the pretraining data? And the more relevant question is how can you assess/confirm that such a sequence has "greater frequency imbalance of these answers in the pretraining data compared to the altered answers"? Specifically without access to the exact pre-training data, such considerations are rough/speculative even if one can argue that there are certain words that we expect to encounter more frequently online and in written texts (e.g. never and always) thus claiming that this would translate necessarily to the pre-training data.

Section 6.2:
- Reddit is generally known to have toxic content. Have the authors done any quality control on the data that they used to form their 1119 responses? Also, while threads where data was scraped from were generally about conversations regarding individuals' personalities, are all samples that make up the dataset actually samples that constitute discussions of personality traits? (e.g. no tangent talk was inadvertently included ?) What quality control has been regarding this issue?

**Reasons To Accept:**

- Interesting study investigating the abilities of LMs to generate personalized text which could be useful to a wide variety of applications (e.g. more personalized chatbots)
- Two new datasets: the first based on collected personality descriptions from human subjects and the second based on personality descriptions collected from Reddit

**Reasons To Reject:**

- The presentation would benefit from some clarifications which I detail in my comments below.
- Concerns regarding the models used in the paper.
- Concerns related to the dataset (see comments below)

**Reproducibility:**

3: Could reproduce the results with some difficulty. The settings of parameters are underspecified or subjectively determined; the training/evaluation data are not widely available.

**Reviewer Confidence:**

4: Quite sure. I tried to check the important points carefully. It's unlikely, though conceivable, that I missed something that should affect my ratings.

**Typos Grammar Style And Presentation Improvements:**

- Throughout the paper (e.g. Line 54, 66, 250 etc), there's language around personality traits of models, I would suggest changing those to personality traits expressed by language models since this is a more accurate description of what you're trying to say.
- Line 93: I'm not familiar with this form "data-set". One could argue that "data set" is correct but nowadays the predominant form is dataset.
- Line 315: A possible explanation for for this
- Line 327: which is* consistent with ...
- Footnote 2:  compared with those without the since

---

> ### Author Rebuttal · Authors · 2023-08-29
>
> We appreciate your time and effort in reviewing our paper. We are grateful for your valuable feedback and thoughtful insights, and we would like to address the concerns you raised:
>
> Responses to Questions:
>
> Question: Line 74: It would be good to clarify here what you mean by context so that the intro is clearer, e.g., you're referring to social contexts? different scenarios where people are interacting with a model for a specific reason, etc...
>
> Response: The word context is used to refer specifically to the text preceding the prompts that the models are evaluated on. In our experiments, we introduce context that comprises of personality descriptions in various forms, such as those based on personality assessment items (Section 6.1), reddit threads (Section 6.2) & self-provided assessments (Section 6.3).
>
> Question: Lines 81-84: I'm not exactly sure what you mean here by "measured personality". Assuming you mean that you qualify the personality expressed by a model based on some psychological assessment, how else would you qualify that expressed personality other than through the generated text by the model? What would be a scenario where the "measured personality" expressed by a model would not be reflected by the text generated by the model?
>
> Response: The “measured personality” is the quantified personality as indicated by the scores from the Big Five personality assessment (Section 5). The generated text might not be reflective of the measured personality if the measured personality was captured inaccurately (i.e. the testing procedure was ineffective). Consider a hypothetical language model that (for some reason) specifically answers psychological questionnaires as an introvert, but generates text and responses to other questions that would match an extrovert. While we do not expect this to happen, it is not clear that it will not, and our experiments provide an answer to this empirical question; experiments in Section 7 confirm that text generated by these models reflect a personality consistent with the measured scores. This provides credibility to the assessment procedure and suggests that the assessment can act as an indicator of personality in text generated from downstream applications.
>
> Question: Lines 261-270: …Big Five assessment has 50 statements/items as shown in the appendix targeting 5 personality traits so there's 10 per trait. You give in this section an example of a "context item" (e.g. in Line 257: I am {blank} the life of the party). So context item is referring to one of those 50 statements/items (which are 10 per trait). Are you then preceding each questionnaire item per trait (e.g. I {blank} feel comfortable around people) by a context item (e.g. I am {blank} the life of the party)? If so, what are you doing when the context item and the questionnaire item are the same? (i.e. you should multiply by 10 or 9 questionnaire items so that you avoid preceding a questionnaire item by the same context item) ? I think this part of the section could benefit from more clarification.
>
> Response: This interpretation of “context item” & “questionnaire item” is generally correct. A “context item” is just a “questionnaire item” used in the assessment context. However, note that the context in this section (6) includes both a “context item” and a modifier (an answer choice used in context). The issue of repeated items is a valid concern. We experimented with an adjustment that accounted for this issue (see line 337). Specifically, we replaced scores for context that consisted of repeat items with the base model score (from Section 5) for the given “questionnaire item”. This meant that the concerning context item could no longer contribute to $\Delta$_{cm}. We saw similar, albeit lower, trends with mean correlations of 0.25 and 0.40, compared with 0.40 and 0.54 for BERT and GPT2 respectively. These correlations remained statistically significant ($p < 0.05$, t-test).
>
>
>
> Alternate framing details: The possible answer choices for testing with the alternate framing are the answer choices from the original, un-modified Big Five assessment, excluding “neutral. These include “agree”, “somewhat agree”, “somewhat disagree” and “disagree”. While we could not determine the exact answer frequency in the pretraining data, analysis with tools like the Google Books ngram viewer gave us some indication of the imbalance in online & literary texts. The generalization of this imbalance to the pretraining data was just one component that factored into the decision to introduce a new framing in Section 4.
>
> Reddit Data Quality: The filtering out of tangent conversations was certainly considered. In the end, no quality control was performed on the data for lack of an objective way to determine how relevant a piece of text was to the topic of personality.
>
> Model choice: BERT and GPT2 were selected because of their availability as open-source models, as opposed to newer models like GPT3. Nonetheless, we agree that this work would benefit from experimentation against the latest language models for the sake of relevancy and accuracy. For instance, as noted in R3, GPT3 handles double negatives better than GPT2. While BERT & GPT2 share similarities to their newer counterparts, further work is required to determine how our results generalize to these other models.
>
> Suggested Citations: Thanks for bringing these to our attention!
>
> General update: We have expanded our discussion of the Big Five taxonomy in Section 2 of the updated paper, as well as the motivation and bases for choosing it above other possibilities such as MBTI and the Enneagram. Our choice is driven by the fact that the Big Five is the most commonly used taxonomy and has been shown to be practically predictive of outcomes such as educational attainment~\cite{o2007big}, longevity and relationship satisfaction. This is not true for any of the alternate taxonomies. Further, it is relatively natural to cast as an assessment for language models. Lastly, we believe that our previous wording in the concluding section painted an unfairly negative picture of Big Five, since the criticisms we mentioned also apply to all other extant personality taxonomies – we have thus removed this offending text.

---

### Official Review · Reviewer_RXZa · 2023-08-08

**Soundness:** 5

**Excitement:**

4: Strong: This paper deepens the understanding of some phenomenon or lowers the barriers to an existing research direction.

**Paper Topic And Main Contributions:**

This paper analyzes the perceived personality traits that are exhibited (if any) by recent language models. Using BERT and GPT2 they extensively analyze if such LMs can account for contextual personality traits and also experiment how such context manifests in the generated output. They further contribute datasets comprising human subjects, which will be of interest to the research community.

**Reasons To Accept:**

1. The analysis presented in this paper is very thorough and the implemented methods are sound.
2. The overall motivation of the paper is strong. The findings of this paper provide significant evidences towards the use of computational methods for estimating personality traits from text.
3. They perform detailed analysis of human feedback, which is crucial for such studies.
4. The curated datasets will be of importance to the community.

**Reasons To Reject:**

1. The generalizability of the findings to other types of computational models is unknown. As a suggestion, it would be interesting if general guidelines could be established that is invariant to the specific model type.

**Reproducibility:**

3: Could reproduce the results with some difficulty. The settings of parameters are underspecified or subjectively determined; the training/evaluation data are not widely available.

**Reviewer Confidence:**

4: Quite sure. I tried to check the important points carefully. It's unlikely, though conceivable, that I missed something that should affect my ratings.

---

> ### Author Rebuttal · Authors · 2023-08-29
>
> We appreciate your time and effort in reviewing our paper. We are grateful for your valuable feedback and thoughtful insights, and we would like to address your concerns:
>
> Model choice: BERT and GPT2 were selected because of their availability as open-source models, as opposed to newer models like GPT3, and ChatGPT had not yet been released when this work was started. Nonetheless, we agree that this work would benefit from experimentation against the latest language models for the sake of relevancy and accuracy. For instance, as noted in R3, GPT3 handles double negatives better than GPT2. While BERT & GPT2 share similarities to their newer counterparts, further work is required to determine how our results generalize to these other models.
>
> General update: We have expanded our discussion of the Big Five taxonomy in Section 2 of the updated paper, as well as the motivation and bases for choosing it above other possibilities such as MBTI and the Enneagram. Our choice is driven by the fact that the Big Five is the most commonly used taxonomy and has been shown to be practically predictive of outcomes such as educational attainment~\cite{o2007big}, longevity and relationship satisfaction. This is not true for any of the alternate taxonomies. Further, it is relatively natural to cast as an assessment for language models. Lastly, we believe that our previous wording in the concluding section painted an unfairly negative picture of Big Five, since the criticisms we mentioned also apply to all other extant personality taxonomies – we have thus removed this offending text.

---

### Official Review · Reviewer_9sqR · 2023-08-09

**Soundness:** 4

**Excitement:**

3: Ambivalent: It has merits (e.g., it reports state-of-the-art results, the idea is nice), but there are key weaknesses (e.g., it describes incremental work), and it can significantly benefit from another round of revision. However, I won't object to accepting it if my co-reviewers champion it.

**Paper Topic And Main Contributions:**

This paper uses psychological questionnaires to test the potential personality of language models and reveals that personality can be manipulated by prefix context in language models. Some empirical analyses show that the personality of human users can be predicted from self-reported text descriptions by language models, and language models can generate text that conforms to human personality.

**Questions For The Authors:**

Will the authors make the two personality description datasets public? This would be beneficial for the community.

**Reasons To Accept:**

1. The insights of this paper are meaningful and helpful in constructing personalized applications.
2. The paper is well-organized.
3. The experiments are detailed and explore different types of contextual data.

**Reasons To Reject:**

1. The lack of results from other personality questionnaire tests, such as Ten Item Personality Inventory (TIPI), makes it unclear whether different questionnaires would lead to different conclusions.
2. The analysis of ChapGPT is missing.
3. It is necessary to clarify the differences between this method and other methods to predict user's personality, such as [1].

[1] Yang F, Yang T, Quan X, et al. Learning to answer psychological questionnaire for personality detection[C]//Findings of the Association for Computational Linguistics: EMNLP 2021. 2021: 1131-1142.

**Reproducibility:**

3: Could reproduce the results with some difficulty. The settings of parameters are underspecified or subjectively determined; the training/evaluation data are not widely available.

**Reviewer Confidence:**

3: Pretty sure, but there's a chance I missed something. Although I have a good feel for this area in general, I did not carefully check the paper's details, e.g., the math, experimental design, or novelty.

---

> ### Author Rebuttal · Authors · 2023-08-29
>
> We appreciate your time and effort in reviewing our paper. We are grateful for your valuable feedback and thoughtful insights, and we would like to address both concerns:
>
> Model choice: BERT and GPT2 were selected because of their availability as open-source models, as opposed to newer models like GPT3, and ChatGPT had not yet been released when this work was started. Nonetheless, we agree that this work would benefit from experimentation against the latest language models for the sake of relevancy and accuracy. For instance, as noted in R3, GPT3 handles double negatives better than GPT2. While BERT & GPT2 share similarities to their newer counterparts, further work is required to determine how our results generalize to these other models.
>
> Suggested Citation: Thanks for bringing this work to our attention. Experiments from [1] use context to influence personally assessment scores in a similar way. However, rather than developing an improved model for predicting measuring, the focus of our work is to 1) characterize personality bias in out-of-the-box, pre-trained large language models and 2) explore ways to manipulate the measured personality (i.e. context). This work, along with a clarification of distinctness off our work, has been added to the literature review in the updated paper.
>
> General update: We have expanded our discussion of the Big Five taxonomy in Section 2 of the updated paper, as well as the motivation and bases for choosing it above other possibilities such as MBTI and the Enneagram. Our choice is driven by the fact that the Big Five is the most commonly used taxonomy and has been shown to be practically predictive of outcomes such as educational attainment~\cite{o2007big}, longevity and relationship satisfaction. This is not true for any of the alternate taxonomies. Further, it is relatively natural to cast as an assessment for language models. Lastly, we believe that our previous wording in the concluding section painted an unfairly negative picture of Big Five, since the criticisms we mentioned also apply to all other extant personality taxonomies – we have thus removed this offending text.
>
> Other: The personality description datasets will be made public.

---

### Official Review · Reviewer_UeN6 · 2023-08-10

**Soundness:** 2

**Excitement:**

3: Ambivalent: It has merits (e.g., it reports state-of-the-art results, the idea is nice), but there are key weaknesses (e.g., it describes incremental work), and it can significantly benefit from another round of revision. However, I won't object to accepting it if my co-reviewers champion it.

**Paper Topic And Main Contributions:**

This study aims to investigate the personality traits of language models. Through experiments and psychological questionnaires, the personality traits of the language models are tested, along with the consistency of their responses and their manipulability.

**Questions For The Authors:**

Based on the definition, it would be more reasonable for the trend depicted by the points in Figure 2 to show that as the absolute values on the x-axis increase, the absolute values on the y-axis also increase？

**Reasons To Accept:**

This paper has conducted sufficient experiments for BERT and GPT2, and the text flows smoothly. It offers insightful inspiration.

**Reasons To Reject:**

The experimental design of this paper is insufficient to demonstrate the manipulability of personality in more powerful language models (GPT3.5..). The evaluation of the experimental design also appears to lack proper justification , as it does not adequately address the biases introduced by the design of prompts and the biases inherent in model parameters. Furthermore, reliability and generalization in the study seems somewhat limited.

**Reproducibility:**

2: Would be hard pressed to reproduce the results. The contribution depends on data that are simply not available outside the author's institution or consortium; not enough details are provided.

**Reviewer Confidence:**

4: Quite sure. I tried to check the important points carefully. It's unlikely, though conceivable, that I missed something that should affect my ratings.

---

> ### Author Rebuttal · Authors · 2023-08-29
>
> We appreciate your time and effort in reviewing our paper. We are grateful for your feedback, and we would like to address the concerns you raised:
>
> Responses to Questions:
>
> Question: Based on the definition, it would be more reasonable for the trend depicted by the points in Figure 2 to show that as the absolute values on the x-axis increase, the absolute values on the y-axis also increase?
>
> Response: The r_{cm} values represent the expected relative change in trait score (expected behavior) when the corresponding context/modifier pair is used as context (see lines 270-279 for an explanation of how this value is calculated). Thus, one would expect X_{cm} evaluated on a more negative r_{cm} to decrease relative to X_{base}. This is to say, we would expect certain context item/modifier pairs to cause a decrease in trait score. For instance, we would expect “I am never the life of the party” to cause a decrease in extroversion score (negative $\Delta$_{cm}). Taking the absolute value of the $\Delta$_{cm} would misrepresent these cases. Cases where the expected & actual behavior is a decrease would instead show as an increase.
>
>
>
> Prompt & Model Biases: The goal was not to eliminate sources of bias, like model parameters & pre-training data, but rather, to see how that bias affects the measured personality of these out-of-the-box models, relative to the population’s measured personality. Further experiments aimed to evaluate how such bias might be accounted for without modifying the model itself (i.e. changing pre-training data & model parameters). This was done by manipulating the models’ behavior via ambient context. Irrespective of the biases in the model parameters, we are interested in exploring the relative change in personality traits of the language models when exposed to textual contexts, rather than their absolute value. We are unclear on your comment about biases in the prompts: our results show that the prompts lead to predictable changes in personality traits. Further our analysis shows that these results are not sensitive to the format of the prompts.

---

### Meta-Review · Area_Chair_TimP · 2023-09-10

**Recommendation:** 3

**Metareview:**

In general, reviews for this paper were mixed, with some reviewers feeling that the paper was more ready for publication than others.  Soundness was rated along a broad range from borderline to excellent, and excitement was rated as ambivalent in most cases.  Overall, the paper could benefit from stronger justification of its arguments and minor to moderate revisions, synthesizing the information (from authors) and feedback (from reviewers) that emerged during the discussion period with the manuscript.  As noted by the authors in their rebuttal, some of these revisions have already been made during the discussion period.

**Summary of Reviewer Feedback and Discussion:**
- **Reviewer UeN6** appreciated the insights arising from the paper's experiments, but felt that the experiments were not designed in such a way that they could demonstrate the manipulability of personality in more powerful LMs.  They also felt that the evaluation of this design was poorly justified and inadequately addressed biases that could be introduced during the experimental process, and they felt that the study's findings had limited reliability and generalizability.  They also raised one question pertaining to a figure.  In their rebuttal, the authors answered this question and clarified the intent of their research, which the reviewer had misunderstood.
- **Reviewer 9sqR** felt that the insights from this paper would be meaningful and helpful for constructing personalized applications and that the experiments were detailed and comprehensive.  However, they felt that it was unclear whether different questionnaires would lead to different conclusions and that an analysis of ChatGPT should have been included.  They also felt that the proposed method could be more comprehensively compared with other personality prediction methods, and asked whether the authors' datasets would be publicly available.  The authors clarified that the datasets would be publicly available, and explained that they had selected open-source models and that ChatGPT had been released after their work started.  They noted that in the future they could also compare to more recent models, and mentioned some general updates that they'd also made to revise the paper.
- **Reviewer RXZa** felt that the paper presented sound methods that were thoroughly analyzed and nicely motivated.  They appreciated the significant findings paired with detailed analyses, and noted that the contributed datasets will be important for the research community.  However, they pointed that the generalizability of the paper's findings to other types of computational models is known, and suggested further studying whether these findings were invariant to model type.  The authors reiterated from their statement to Reviewer 9sqR that further work is needed to asses generalizability, and also mentioned their general updates to the manuscript.
- **Reviewer gkMG** felt that the study was interesting and may be useful to numerous applications, and also appreciated the new datasets introduced in the paper.  However, they raised many points requiring clarification, and some concerns regarding the models used in the paper and the datasets themselves.  Specifically, they felt that the selected models were too old, raising concerns that the paper's conclusions may not carry over to more recent models, and they were concerned whether adequate quality control was performed during the dataset development process.  The authors provided clarifying responses to Reviewer gkMG's questions, and reiterated earlier points made to other reviewers about their selection of open-source models and the need for further studies to assess broader generalizability.  They noted that no quality control was performed since there was no objective way to determine text relevance to the topic of personality.  Reviewer gkMG thanked them for their rebuttal and clarified that their interest in quality control mainly pertained to toxic language, and suggested that if no quality control was performed, then a qualitative analysis should be performed on a random sample of the data as a post-hoc quality evaluation.  The authors agreed that this would be helpful, and they promised to add this in the final version of the paper.

---

### Decision · Program_Chairs · 2023-10-07

**Decision:**

Accept-Findings

**Comment:**

In general, reviews for this paper were mixed, with some reviewers feeling that the paper was more ready for publication than others.  Soundness was rated along a broad range from borderline to excellent, and excitement was rated as ambivalent in most cases.  Overall, the paper could benefit from stronger justification of its arguments and minor to moderate revisions, synthesizing the information (from authors) and feedback (from reviewers) that emerged during the discussion period with the manuscript.  As noted by the authors in their rebuttal, some of these revisions have already been made during the discussion period.

**Summary of Reviewer Feedback and Discussion:**
- **Reviewer UeN6** appreciated the insights arising from the paper's experiments, but felt that the experiments were not designed in such a way that they could demonstrate the manipulability of personality in more powerful LMs.  They also felt that the evaluation of this design was poorly justified and inadequately addressed biases that could be introduced during the experimental process, and they felt that the study's findings had limited reliability and generalizability.  They also raised one question pertaining to a figure.  In their rebuttal, the authors answered this question and clarified the intent of their research, which the reviewer had misunderstood.
- **Reviewer 9sqR** felt that the insights from this paper would be meaningful and helpful for constructing personalized applications and that the experiments were detailed and comprehensive.  However, they felt that it was unclear whether different questionnaires would lead to different conclusions and that an analysis of ChatGPT should have been included.  They also felt that the proposed method could be more comprehensively compared with other personality prediction methods, and asked whether the authors' datasets would be publicly available.  The authors clarified that the datasets would be publicly available, and explained that they had selected open-source models and that ChatGPT had been released after their work started.  They noted that in the future they could also compare to more recent models, and mentioned some general updates that they'd also made to revise the paper.
- **Reviewer RXZa** felt that the paper presented sound methods that were thoroughly analyzed and nicely motivated.  They appreciated the significant findings paired with detailed analyses, and noted that the contributed datasets will be important for the research community.  However, they pointed that the generalizability of the paper's findings to other types of computational models is known, and suggested further studying whether these findings were invariant to model type.  The authors reiterated from their statement to Reviewer 9sqR that further work is needed to asses generalizability, and also mentioned their general updates to the manuscript.
- **Reviewer gkMG** felt that the study was interesting and may be useful to numerous applications, and also appreciated the new datasets introduced in the paper.  However, they raised many points requiring clarification, and some concerns regarding the models used in the paper and the datasets themselves.  Specifically, they felt that the selected models were too old, raising concerns that the paper's conclusions may not carry over to more recent models, and they were concerned whether adequate quality control was performed during the dataset development process.  The authors provided clarifying responses to Reviewer gkMG's questions, and reiterated earlier points made to other reviewers about their selection of open-source models and the need for further studies to assess broader generalizability.  They noted that no quality control was performed since there was no objective way to determine text relevance to the topic of personality.  Reviewer gkMG thanked them for their rebuttal and clarified that their interest in quality control mainly pertained to toxic language, and suggested that if no quality control was performed, then a qualitative analysis should be performed on a random sample of the data as a post-hoc quality evaluation.  The authors agreed that this would be helpful, and they promised to add this in the final version of the paper.